# An optimal posttreatment surveillance strategy for cancer survivors based on an individualized risk-based approach

Guan-Qun Zhou[1,2,5], Chen-Fei Wu[3,5], Bin Deng[4,5], Tian-Sheng Gao[4], Jia-Wei Lv[1,2], Li Lin[1,2], Fo-ping Chen[1,2], Jia Kou[1,2], Zhao-Xi Zhang[3], Xiao-Dan Huang[1,2], Zi-Qi Zheng[1,2], Jun Ma[1,2], Jin-Hui Liang[4✉] & Ying Sun [1,2✉]

The optimal post-treatment surveillance strategy that can detect early recurrence of a cancer within limited visits remains unexplored. Here we adopt nasopharyngeal carcinoma as the study model to establish an approach to surveillance that balances the effectiveness of disease detection versus costs. A total of 7,043 newly-diagnosed patients are grouped according to a clinic-molecular risk grouping system. We use a random survival forest model to simulate the monthly probability of disease recurrence, and thereby establish risk-based surveillance arrangements that can maximize the efficacy of recurrence detection per visit. Markov decision-analytic models further validate that the risk-based surveillance outperforms the control strategies and is the most cost-effective. These results are confirmed in an external validation cohort. Finally, we recommend the risk-based surveillance arrangement which requires 10, 11, 13 and 14 visits for group I to IV. Our surveillance strategies might pave the way for individualized and economic surveillance for cancer survivors.

[1] State Key Laboratory of Oncology in South China, Collaborative Innovation Center for Cancer Medicine, Guangdong Key Laboratory of Nasopharyngeal Carcinoma Diagnosis and Therapy, Guangzhou, China. [2] Department of Radiation Oncology, Sun Yat-sen University Cancer Center, 510060 Guangzhou, China. [3] Zhongshan School of Medicine, Sun Yat-sen University, 510060 Guangzhou, China. [4] Department of Radiation Oncology, Wuzhou Red Cross Hospital, Guangzhou 543002 Guangxi, China. [5] These authors contributed equally: Guan-Qun Zhou, Chen-Fei Wu, Bin Deng. ✉email: 566jv@sina.cn; sunying@sysucc.org.cn

Oncological surveillance practice guidelines, based largely upon expert opinion, recommend that cancer survivors receive regular posttreatment surveillance (i.e., repeated physical, hematological, and radiological examinations averagely every 3–6 months) to facilitate early detection of disease recurrence[1–3]. This strategy has been proved to be effective to improve prognosis in multiple cancer types, as early detected recurrent lesions are more responsive to salvaged therapies. However, undertaking this surveillance strategy on a consistent scale seems to be burdensome for the patients, given their heterogeneity risks and patterns of recurrence. In addition, it requires intensive global medical sources. Currently, question remains that what is the optimal posttreatment surveillance strategy that could detect recurrence in a timely manner and be cost-effective.

Therefore, we aim to address this evidence gap by establishing an individualized, risk-based surveillance strategy in a large-scale cohort. We take nasopharyngeal carcinoma (NPC) as the study model as: (i) NPC is endemic in Southeast Asia, with age-standardized rate of 3.0 per 100,000 in China;[4] therefore, cost-effective surveillance strategy is essentially important to reduce patient costs in those developing countries. (ii) Our cancer center treats more than 4000 newly-diagnosed NPC patients per year. We have established a NPC disease-specific big-data intelligence research platform since 2015[5], which included over 50,000 NPC cases with high-quality follow-up information; therefore, greatly facilitated this large-scale, population-based research. (iii) Nasopharynx cancer is sensitive to radiotherapy and chemotherapy; early-stage recurrent disease or oligometastatic lesions were able to achieve favorable survival outcomes[6]. (iv) The National Comprehensive Cancer Network (NCCN) for head and neck cancer recommended uniform follow-up strategy across all NPC patients[7], whereas survival outcomes of this disease were heterogeneous and significantly varied among different stages. The 5-year overall survival varied from 65% for locally advanced disease to 90% for early stage disease[8], and were further sub-stratified by levels of molecular biomarker (Epstein–Barr virus Deoxyribo Nucleic Acid [EBV DNA])[9].

In this study, we adopt a random survival forest (RSF) model to simulate the monthly probability of disease recurrence, and further establish risk-based surveillance arrangements based on the recurrence probability algorithm. Surveillance arrangements are evaluated using the sum of delayed-detection months and cost-effectiveness analysis, and compared with the NCCN surveillance guidelines. Although the model is established in the context of NPC, our method of modeling risk-based surveillance may be applicable for the development of cost-effective surveillance strategies for other diseases, and could assist in shaping individualized, risk-based posttreatment follow-up for the cancer survivors in general.

## Results

**Patient characteristics and follow-up status**. In total, 6416 patients in the training cohort and 627 patients in the validation cohort were identified eligible in this study. Patient baseline demographic and disease features were summarized in Table 1. All patients were followed-up according to the institutional follow-up guidelines. In the training cohort, after a median follow-up of 61.8 month (IQR, 52.1–73.7 months), there were a total of 56,782 routine visits for the whole cohort (median, 10; IQR, 5–14). As for regular imaging examinations, a total of 21,823 surveillance magnetic resonance imaging (MRI) or computed tomography (CT) of the nasopharynx, 22,932 MRI, CT, or sonography of the neck, 18,026 abdominal CT or sonography, 17,119 chest X-ray or CT, and 8552 ECT were performed.

**Incidences and patterns of failure**. In the training cohort, we recorded 428 (6.7%) local recurrence, 358 (5.6%) regional

**Table 1 Demographic and baseline characteristics of patients in the training and validation cohorts.**

| Characteristics | Training cohort (N = 6416) | Validation cohort (N = 627) |
|---|---|---|
| Age, years | | |
| <45 | 3116 (48.6) | 203 (32.3) |
| ≥45 | 3300 (51.4) | 424 (67.7) |
| Gender | | |
| Male | 4753 (74.1) | 446 (71.1) |
| Female | 1663 (25.9) | 181 (28.9) |
| Smoking | | |
| No | 4107 (64.0) | 520 (82.9) |
| Yes | 2309 (36.0) | 107 (17.1) |
| Alcohol | | |
| No | 5510 (85.9) | 549 (87.6) |
| Yes | 906 (14.1) | 78 (12.4) |
| Family history | | |
| No | 4706 (73.3) | 603 (96.2) |
| Yes | 1710 (26.57) | 24 (3.8) |
| Histological type | | |
| WHO Type I | 36 (0.6) | 25 (4) |
| WHO Type IIa/IIb | 6380 (99.4) | 602 (96) |
| T category[a] | | |
| T1 | 667 (10.4) | 96 (15.3) |
| T2 | 1141 (17.8) | 146 (23.3) |
| T3 | 3048 (47.5) | 141 (22.5) |
| T4 | 1560 (24.3) | 244 (38.9) |
| N category[a] | | |
| N0 | 820 (12.8) | 43 (6.9) |
| N1 | 2965 (46.2) | 326 (52.0) |
| N2 | 1973 (30.8) | 170 (27.1) |
| N3 | 658 (10.3) | 88 (14.0) |
| EBV DNA | | |
| ≤2000 copies/mL | 2971 (46.3) | 468 (74.6) |
| >2000 copies/mL | 3445 (53.7) | 159 (25.4) |
| Groupings[b] | | |
| I | 367 (5.7) | 22 (3.5) |
| II | 3472 (54.1) | 300 (47.8) |
| III | 1863 (29.0) | 217 (34.6) |
| IV | 714 (11.1) | 88 (14.0) |
| Chemotherapy | | |
| CRT | 5678 (88.5) | 589 (93.9) |
| RT alone | 738 (11.5) | 38 (6.1) |
| HGB | | |
| ≤130 g/L | 1180 (18.4) | 270 (43.1) |
| >130 g/L | 5236 (81.6) | 357(56.9) |
| ALB | | |
| ≤40 g/L | 609 (9.5) | 161 (25.7) |
| >40 g/L | 5807 (90.5) | 466 (74.3) |
| CRP | | NA |
| ≤3 mg/L | 6515 (69.7) | |
| >3 mg/L | 2833 (30.3) | |
| LDH | | NA |
| ≤245 IU/L | 5,891 (91.8) | |
| >245 IU/L | 515 (8.2) | |

WHO, World Health Organization; EBV, Epstein–Barr virus; HGB, hemoglobin; ALB, albumin; CRP, C-reactive protein; LDH, lactate dehydrogenase; CRT, chemoradiotherapy; RT, radiotherapy; IU, international unit (s).
[a]According to the American Joint Committee on Cancer, 8th edition.
[b]Patients were grouped according to T category, N category and EBV DNA.

recurrence, and 854 (13.3%) distant metastasis events, with median durations of 27.9, 25.0, and 18.7 months, respectively. The distribution of disease failure events was detailed in Supplementary Fig. S1. The number of disease failure events in each month was counted for group I–IV (Supplementary Fig. S1A). To eliminate the impact of different patient number in each group, crude disease failure incidence was calculated using the events

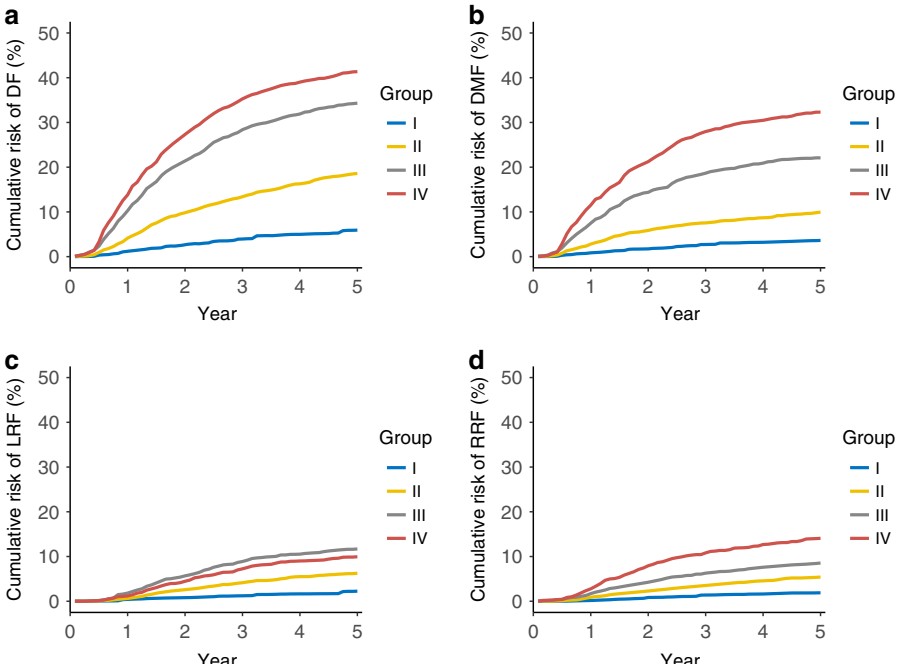

**Fig. 1 Risk-adjusted survival curves of different staging groups in nasopharyngeal carcinoma patients.** Disease failure probabilities (**a**); distant metastasis–free survival probabilities (**b**); local recurrence–free survival probabilities (**c**) and regional recurrence–free survival probabilities (**d**). Number of patients: group I, $n = 367$; group II, $n = 3472$; group III, $n = 1863$; group IV, $n = 714$.

number divide by the total patients at risk (Supplementary Fig. S1B).

**Risk-adjusted disease failure probability**. To calculate the risk of disease failure of the four risk groups in each month, we used RSF to simulate the monthly probability of disease failure, adjusted for the well-established risk factors of NPC (Supplementary Fig. S2). The RSF-simulated recurrence probabilities of patients stratified by risk groupings were illustrated in Fig. 1; 5-year risk-adjusted DF probabilities for group I to IV were 5.4%, 18.6%, 34.3%, and 41.4%, respectively.

Next, we calculated the time-specific event-occurring probability for disease recurrence (Fig. 2). For patients in group I, it was demonstrated that the disease failure incidence was extremely low, and the curves for all endpoints were relatively flat with no prominent peak. For patients with group II, we identified miniature peaks for metastasis within 2 years, but not for local/regional recurrence. Whereas, for patients with more advanced disease (in group III–IV), we observed remarkable peaks for all endpoints.

The pattern of peak occurrence time varied in different endpoints. For disease failure, the peak occurrence time was around 15 month after treatment, and this trend was consistent in all patients of group II–IV. Generally, the peak occurrence of distant metastasis was earlier than that of local and regional recurrence. Distant metastasis occurred mainly around 12th to 15th month, and delayed distant metastases after the first three years were relatively rare. As for local and regional recurrence, the peak occurrence time was approximately the 18th to 24th month, and there were still a considerable number of recurrences in the 4–5 years after treatment.

**Establishment of risk-based surveillance strategy**. Then, we established risk-based surveillance arrangements according to monthly disease failure probability using the predetermined rules. Figure 3 shows the risk-based arrangement for different total

number of follow-ups (varying from 27 to 5, the abscissa in Fig. 3) from year 1 to 5 for groups I–IV (the ordinate in Fig. 3). The follow-up arrangements of a total of 13 visits in patients in group III (4, 4, 3, 1 and 1 visits in years 1–5 respectively) were highlighted in the revised Fig. 3 for clarify: we scheduled the follow-up time points (the 7th, 8th, 10th, 12th, 14th, 16th, 19th, 23rd, 27th, 30th, 36th, 44th and 52nd month post-IMRT) for surveillance according to pre-established algorithms (Supplementary Table S1).

Consistent with the distributions of risk-adjusted disease failure probabilities, the risk-based surveillance arrangements were relatively averaged in patients with earlier disease, manifesting considerable number of follow-up in year 4–5. On contrast, in patients with advanced disease, the risk-based surveillance arrangements mainly concentrated in the first 3 years after treatment, relatively fewer follow-up was assigned to the subsequent years.

**Oncologic surveillance performance comparison**. The oncologic surveillance performance (ability to detect disease failure timely) of risk-based surveillance arrangements were compared with that of control strategies (Fig. 4). As we seen, with the total number of follow-ups declining from 27 visits to 5 visits, the relevant delayed-detection time was gradually increasing. Delayed-detection time of risk-based surveillance arrangements (blue curve) was compared with those of the control strategies (the yellow, red, and grass green point represented the most, moderately and least intensive surveillance NCCN strategies; the light green point represented the RTOG strategy). As we seen, at the same number of visits, the risk-based surveillance arrangements reduced the delayed-detection time substantially, and it is obviously more effective than the control strategies.

In patients with earlier disease, the advantages of risk-based surveillance arrangements were more significant. For patients of group I, the disease detection ability of risk-based 12 visits arrangement was superior to that of 27 times according to most intensive NCCN surveillance, and the ability of risk-based ten times equaled that of 14–15 visits according to the moderately intensive NCCN and RTOG strategies (Fig. 4a). In patients of

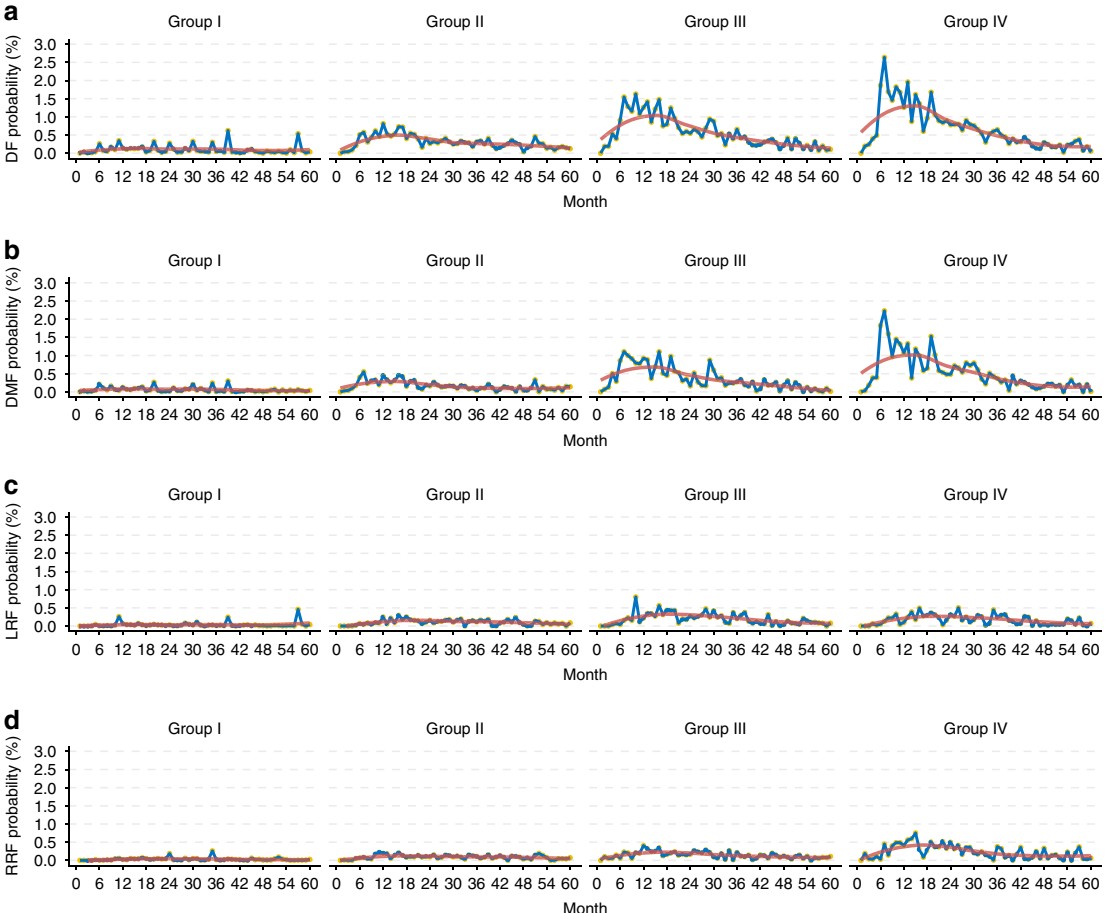

**Fig. 2 Time-specific occurrence probabilities of different staging groups in nasopharyngeal carcinoma patients.** Disease failure (**a**); distant metastasis (**b**); local recurrence (**c**) and regional recurrence (**d**). Number of patients: group I, $n = 367$; group II, $n = 3472$; group III, $n = 1863$; group IV, $n = 714$.

group II, the effectiveness of risk-based surveillance arrangements was also significant, with 11 visits equal to that of 14–15 visits according the moderately intensive NCCN and RTOG strategies (Fig. 4b). As for patients in group III and IV, the superiority of the risk-based surveillance was reduced. Risk-based arrangement of 13 visits and 14 visits for group III and IV, respectively has only slight advantages on that of 15 and 14 visits according to the NCCN and RTOG strategies (Fig. 4c, d).

**Cost effectiveness analysis.** The baseline cost-effectiveness analyses were summarized in Table 2. In general, the risk-based surveillance arrangement was superior to the control strategies in all the groups. For patients in group I, the ICERs for the moderately intensive NCCN surveillance, the most intensive NCCN surveillance, the RTOG strategy and the risk-based strategy were $2848/QALY, $5072/QALY, $2800/QALY, and $1957/QALY, respectively, when compared with the least intensive NCCN strategy. Thus, the risk-based strategy had the most favorable ICER. For the remaining patients groups, the risk-based strategy was also the most cost-effective, with ICERs gradually increasing from group II to IV ($2407/QALY, $3070/QALY, and $3229/QALY, respectively).

Cost effectiveness analysis in the validation cohort showed that the risk-based strategy was dominant compared with the control strategies in all groups, and the baseline results are summarized in Supplementary Table S2. Although the demographic and baseline characteristics of the validation cohort were rather different from those of the training cohort, the risk-based surveillance strategy showed excellent applicability.

**Recommendation for risk-based surveillance strategies.** Referring to the efficiency of the moderately intensive NCCN and RTOG strategies, we recommended risk-based surveillance arrangements for patients in different groups (Fig. 5). For patients of group I, the risk-based surveillance arrangements were total ten visits within 5 years (2, 3, 2, 2, and 1, visits respectively in year 1–5). For patients in group II, it has 11 visits (2, 4, 2, 2, and 1 visits respectively in year 1–5). We recommended 13 visits (4, 4, 3, 1, and 1 visits, respectively) for patients in group III and 14 visits for group IV (4, 5, 3, 1, and 1 visits, respectively). The specific timing for each visit was shown in Fig. 5.

The pattern of distant failure was almost consistent for all groups. Monitoring of distant metastasis should be focused on the first 3 years after treatment. Generally, the occurrence of local and regional recurrence was later, and the monitoring of locoregional recurrence should be focused on year 2–3 after treatment for patients in group I and II. While for patients in group III–IV, it should be continuous in year 2–5 after treatment.

## Discussion

Considering the disease failure potential after treatment, ongoing surveillance was essential for cancer survivors. However, limited data was available regarding the optimal follow-up schedule. In this large and comprehensive population-based study, we created an approach using RSF to qualify the monthly risk of disease failure. Based on these, we developed a risk-based surveillance schedule which could maximize events detected per visit, and it's more economic and efficient compared with surveillance strategies recommended by the NCCN and RTOG.

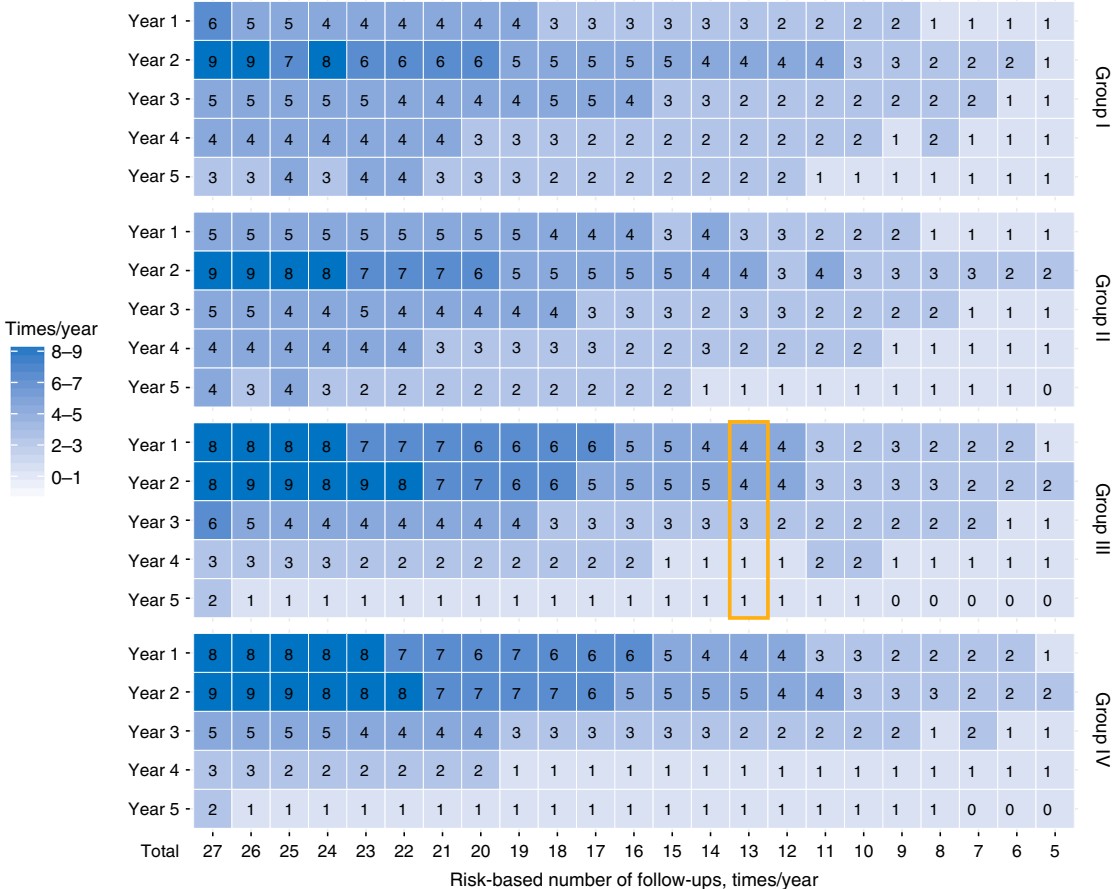

**Fig. 3 The risk-based surveillance arrangements varied 27 to 5 visits for early detection of disease failure.** The follow-up arrangements for a total of 13 visits for group III are highlighted in the orange box.

A number of professional organizations have proposed guidelines for the management after the primary treatment, and universally provided follow-up recommendations for all patients. Take NPC for example, most of these recommendations were derived from those used for non-nasopharyngeal squamous cell carcinoma of head and neck. So the surveillance experience may not applicable to NPC patients, as the biological behavior, treatment mode, and recurrence pattern differed substantially. On the other hand, there was no definitive evidence to clarify whether the surveillance guidelines was efficacious. As a result, significant heterogeneity in the delivery of surveillance care has developed, leading to both over- and under-utilization of testing for certain patient groups[10].

If the main objective of oncologic surveillance were to detect events as early as possible, the ideal situation would be to examine the patient as frequently as possible, daily or weekly. Only in this way it would be possible to detect any event infinitely proximity to the true time of its occurrence. Obviously this ideal situation was not economically nor practically feasible. Surveillance strategies could nevertheless be optimized according to the tumor failure probability, by placing follow-up closer to the times when failures are expected to occur. It was essential for effective and efficient patient care, both medically and financially.

In our analysis, the risk-based surveillance schedule could maximize events detected per visit, and it's more economic and efficient compared with surveillance strategies recommended by the NCCN and RTOG. Considering the cost and feasibility, we choose the moderately intensive NCCN and RTOG strategies as a reference. In fact, physicians could choose any more effective or more economical follow up regimen, depending on the patient's requirements and economic conditions.

The timely detection of local and regional recurrence events was usually deemed extremely essential on the oncologic surveillance, because many early local recurrent lesions can be effectively treated. For NPC, since the 5-year overall survival rates in early recurrences patients was as high as 75%, while the 5-year survival for patients with advanced recurrence disease was only 35%[6,11,12]. In our analysis, the occurrence peak of local and regional recurrences occurred in the 18th to 24th months approximately, and there were still a considerable number of recurrences in the 4–5 years after treatment. This finding is consistent with previous data by Lee AW et al, who reported the proportion of recurrence with latency <2 years, 2-<5 years, and 25 years were 52%, 39%, and 9%, respectively[13]. In another study from Liu X et al more recently, they reported the hazard curve of disease failure at 2 years from commencement of primary radiotherapy[8]. Therefore, the follow-up of primary and local lesions should be focused on the second year after treatment, and follow-up should continue indefinitely in year 3–5 because late recurrence can occur.

As for the surveillance modality for local and regional recurrence, the most commonly used method in clinical practice is H&P (history and physical examination, including neck palpation and nasopharyngoscopy) or imaging examination (preferably MRI). Given the fairly favorable rate of locoregional control achieved after radical IMRT for NPC patients[14], imaging examination for suspicious patients after initial screening by H&P might be a viable option. According to our cost-effectiveness analysis, routine MRI surveillance for the head and neck was not cost-effective due to the high cost of MRI coupled with low rates of failure in T1–2 patients, while annual

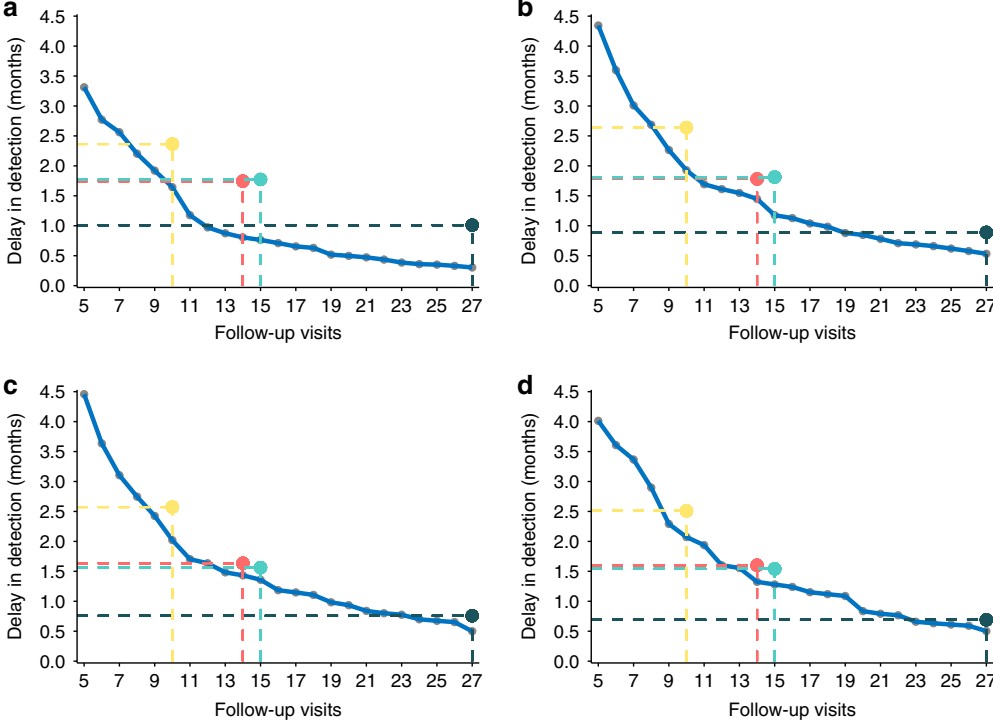

**Fig. 4 Comparisons of delays in the detection of disease failure in different surveillance strategies for each staging group.** Delays in the detection of disease failure in risk-based surveillance arrangements (blue curve) compared with the control follow-up strategies (the yellow, green, and blue horizontal lines, respectively, represent the most intensive, moderately intensive and least intensive surveillance strategies according to NCCN; the brown horizontal line represents the RTOG strategy) for patients in group I (**a**), group II (**b**), group III (**c**), and group IV (**d**). Note that the delays in detection was calculated using a hypothetical cohort of 1000 nasopharyngeal carcinoma patients with the same features with our study population.

MRI surveillance was the dominant and possibly a cost-effective strategy for T3–4 patients[15].

Due to the improved locoregional control, distant metastasis has become a predominant pattern of treatment failure for NPC patients[16]. According to our data, the occurrence of distant metastasis was much earlier than that of locoregional recurrence, and delayed distant metastases were relatively rare after the first 3 years. Our result was similar to a large NPC cohort treated with 2-dimensional radiotherapy, which reported the median remission from the end of radiation to metastasis of 13 months[17]. In another recent study based on IMRT data, the median duration was 15 months. Cumulative metastatic rates at 1-, 2-, and 3-years were 38.5%, 70.3%, and 95.6%, respectively[18]. These results indicated that it may be necessary to closely monitor for distant metastasis within 2 years after treatment, and less intensive follow-up for distant metastases after 3 years was relatively safe.

As for the follow up modality for distant metastasis, advanced metastasis disease can be detected by patient symptoms or physical examination results, while early detection of the vast majority of mild metastatic disease depends mainly on imaging. Although NPC with distant metastasis was usually considered incurable[19], early detection and treatment of isolated asymptomatic disease could improve survival, with the 3 years survival rate of ~60% in patients with isolated metastatic lesion, while the survival rate is only 30% in patients with multiple lesions[20,21]. Therefore, early diagnosis of metastatic NPC via routine body imaging instead of symptoms may be of great clinical value.

Several limitations of our study should be emphasized. Except for the surveillance of tumor recurrence, the clinical follow-up care for cancer survivors also include screening for second primary cancers, assessment and management of long-term and late adverse events, health promotion, and care coordination[22]. In this study, we mainly concentrate on the survival endpoints, as they

are the top concerns by cancer patients that ought to be detected in a cost-effective manner; other unurgent evaluations can be carried out together with the oncologic surveillance. Secondly, cancer survivors need to be followed up lifelong posttreatment, while our analyses mainly focus on the first 5 years post-radiotherapy. Nonetheless, according to our previous study, more the 90% of disease failure occurred within the first 5 years[8]. Lastly, we only took in two strongest prognostic indicators (the clinical stage and EBV DNA) for patient grouping. As an alternative, we entered the other factors into the random survival forest model for risk adjustment.

In this population-based study, we created an approach using RSF to qualify the monthly risk of disease failure for cancer survivors. Basing the frequency of follow-up on disease failure probability, we developed a risk-based surveillance schedule which could maximize events detected per visit. The recommended surveillance strategy was more economically effective compared with compared with the NCCN and RTOG surveillance strategies.

## Methods

**Patient population and data extraction.** The study cohort were comprised of 7043 histologically proven, non-disseminated NPC patients from two academic institutions. The training cohort was comprised of 6416 patients diagnosed between 2009 and 2014, identified from a NPC-specific database from the well-established big-data intelligence platform at Sun Yat-Sen University Cancer Centre (SYSUCC). It is a big-data research system that enables real-time organizing, linking, and structuring data from a number of clinical business systems and thus allows health care providers to perform multidimensional big-data queries[5]. An external cohorts from the Wuzhou Red Cross Hospital (WZRCH) were enrolled for validation (N = 627, diagnosed between 2012 and 2015). The authenticity of this study has been validated by uploading the key raw data onto the research data deposit (RDD) public platform (http://www.researchdata.org.cn), with the approval RDD number as RDDA2018000934. The institutional ethics committees of SYSUCC and WZRCH approved the study protocol and waived the requirement

**Table 2 Baseline cost-effectiveness analysis in Markov models.**

| | Cost ($) | Incremental cost ($) | Effectiveness(QALYs) | Incremental effectiveness | ICER ($/QALY) |
|---|---|---|---|---|---|
| Patients in group I[a] | | | | | |
| The least intensive NCCN strategy | 9187 | 0 | 36.049 | 0 | 0 |
| The moderately intensive NCCN strategy | 11,138 | 1951 | 36.734 | 0.685 | 2848 |
| The most intensive NCCN strategy | 15,699 | 6512 | 37.333 | 1.284 | 5072 |
| The RTOG strategy | 11,273 | 2050 | 36.780 | 0.732 | 2800 |
| The risk-based strategy[b] | 9372 | 185 | 36.142 | 0.093 | 1957 |
| Patients in group II[a] | | | | | |
| The least intensive NCCN strategy | 12,479 | 0 | 26.627 | 0 | 0 |
| The moderately intensive NCCN strategy | 15,298 | 2819 | 27.596 | 0.969 | 2909 |
| The most intensive NCCN strategy | 19,911 | 7432 | 28.288 | 1.661 | 4474 |
| The RTOG strategy | 15,732 | 3253 | 27.734 | 1.107 | 2939 |
| The risk-based strategy[b] | 14,869 | 2390 | 27.620 | 0.993 | 2407 |
| Patients in group III[a] | | | | | |
| The least intensive NCCN strategy | 14,815 | 0 | 21.626 | 0 | 0 |
| The moderately intensive NCCN strategy | 17,821 | 3006 | 22.514 | 0.888 | 3385 |
| The most intensive NCCN strategy | 22,135 | 7320 | 23.040 | 1.414 | 5177 |
| The RTOG strategy | 18,333 | 3518 | 22.690 | 1.064 | 3306 |
| The risk-based strategy[b] | 17,864 | 3049 | 22.619 | 0.993 | 3070 |
| Patients in group IV[a] | | | | | |
| The least intensive NCCN strategy | 15,970 | 0 | 20.264 | 0 | 0 |
| The moderately intensive NCCN strategy | 19,111 | 3141 | 21.146 | 0.882 | 3561 |
| The most intensive NCCN strategy | 23,367 | 7397 | 21.645 | 1.381 | 5356 |
| The RTOG strategy | 19,546 | 3576 | 21.317 | 1.053 | 3396 |
| The risk-based strategy[b] | 19,564 | 3594 | 21.377 | 1.113 | 3229 |

QALY, quality-adjusted life years; ICER, incremental cost-effectiveness ratio; NCCN, National Comprehensive Cancer Network; RTOG, Radiation Therapy Oncology Group.
[a]Patients were grouped according to TNM stages and EBV DNA.
[b]The dominant strategy.

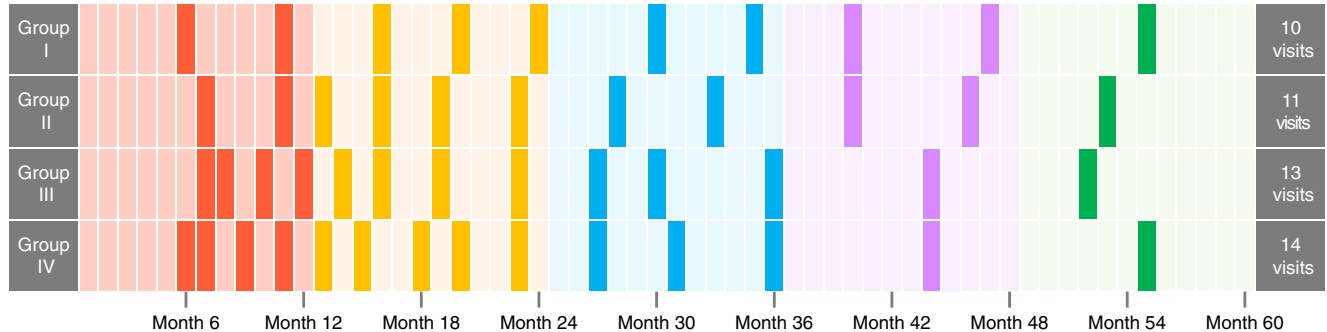

**Fig. 5 Recommended risk-based surveillance arrangements for patients in each group from year 1 to 5.** The darkened boxes of the grid represent the months recommended for visits.

for informed consent given the retrospective nature of the study (No. B2020-138-01 and No. LL2020-37).

All patients were treated with radical intensity modulated radiation therapy (IMRT) comprising five daily fractions delivered per week for 6–7 weeks[23]. Overall, 89.3% patients received chemotherapy.

**Pretreatment evaluations, IMRT techniques, and chemotherapy**. All patients underwent pretreatment evaluations, including a complete history, physical examination, hematology and biochemistry profiling, fiberoptic nasoendoscopy, MRI or CT of the head and neck, chest radiography or CT, abdominal ultrasonography, whole body bone scan (ECT), or 18F-Fluorodeoxyglucose positron emission tomography and computed tomography (PET-CT).

The prescribed doses were 66–72 Gy/28–33 fractions to the planning target volume (PTV) of the primary gross tumor volume, 64–70 Gy/28–33 fractions to the PTV of the GTV of the involved lymph nodes, 60–63 Gy/28–33 fractions to the PTV of the high-risk clinical target volume (CTV1), and 54–56 Gy/28–33 fractions to the PTV of the low-risk clinical target volume (CTV2).

Institutional guidelines recommended radical IMRT for patients with stage I, platinum-based concurrent chemoradiotherapy ± introduction chemotherapy/adjuvant chemotherapy for stage II to IVB NPC. Reasons for deviation from the guidelines included recruitment in clinical trials, individual patient's refusal, age, or organ dysfunction suggestive of intolerance to treatment. When possible, salvage treatments were provided for patients with relapse or persistent disease.

**Patient groupings**. All patients underwent pretreatment evaluations, including a complete history, physical examination, hematology and biochemistry profiling, fiberoptic nasoendoscopy, MRI or CT of the head and neck, chest radiography or CT, abdominal ultrasonography, whole body bone scan (ECT), or 18F-Fluorodeoxyglucose positron emission tomography and computed tomography. Plasma EBV DNA concentrations were measured by real-time quantitative polymerase chain reaction assay of the BamHI-W region of the EBV genome[22]. Details of EBV DNA quantification are described in the Supplementary Methods.

Patients were restaged by two radiation oncologists specializing in head and neck cancer according to the 8th edition of the American Joint Commission on Cancer (AJCC) staging system[24]. Then patients were grouped based on T category, N category and EBV DNA according to a previous study in that it demonstrated better prognostic performance than the AJCC staging system:[25] group I (T1N0), group II (T2-3N0 or T1-3N1; T1-3N2, EBV DNA ≤ 2000 copies/mL), group III (T1-3N2, EBV DNA > 2000 copies/mL; T4N0-2), and group IV (any T and N3).

**Posttreatment follow-up.** Pre-treatment baseline imaging of the primary (nasopharynx, neck) and distant sites (chest, abdomen, bone) were repeated within the 3 months after treatment, and then patients were followed up every 3 months during the first 2 years, every 6 months for year 3–5, and then annually thereafter. During every visit, a complete history, clinical examinations, nasopharyngoscopy, and plasma EBV DNA were routinely performed. Repeating imaging of the primary and distant sites was arranged at least once a year. Patients with clinical suspicion of disease failure were recommended for MRI, CT, abdominal sonography, ECT or even PET/CT, followed by confirmatory cytological biopsies if possible. The follow-up duration was calculated from the end of radiotherapy to the day of death or the last examination.

**Establishment of risk-based surveillance strategy.** To identify the optimal surveillance strategy, a two-stage algorithm was established. At the first stage, we depicted the cumulative probabilities of disease recurrence over-time for patients at different risk groups (defined by NPC clinic-molecular staging system) using the random survival forest (RSF) model, and calculated the monthly time-specific, event-occurring probabilities. Next, we set the total number of visits from the least 5 times (followed-up every per year) to the most 27 times (followed-up according to NCCN guideline). The follow-up visits were allocated to every month based on the monthly recurrence probability, which was defined as any type of recurrence event, including local recurrence, regional recurrence and distant metastasis. The surveillance strategy was evaluated using the sum of delayed-detection months, and compared to the effectiveness of NCCN and Radiation Therapy Oncology Group (RTOG) surveillance guidelines. The best strategy was chosen if it could detect disease recurrence in a timely manner with the least visit times. Detailed statistical considerations were described as follows.

The recurrence probabilities of the following end points (time to the first defining event) were assessed: disease failure (DF), distant metastasis failure (DMF), local recurrence failure (LRF), and regional recurrence failure (RRF). DF was calculated from the date of diagnosis to the date of local or regional recurrence, distant metastasis, or death from any cause, whichever occurred first. DMF, LRF, and RRF were calculated from the date of diagnosis to the dates of distant metastasis, local recurrence and regional recurrence, respectively. To fill the missing value of risk in months that no disease failure occurs, as well to reduce the bias due to unbalanced actual follow-up time of retrospective population, RSF was used to simulate the probability of disease failure in each month.

RSF is an ensemble learning method for the analysis of right censored survival data[26]. RSF methodology produced a survival curve for each group of patients that was risk-adjusted for all variables including sex, age, smoke, alcohol, family history, EBV DNA, HGB, ALB, CRP, LDH, T stage, N stage[27–30]. Briefly, a forest of 1000 random bootstrap survival trees was grown using log-rank splitting for each patient. On average, each tree was grown from 63% of the data; the remaining 37% data are referred to as out-of-bag (OOB) data. Each tree and its corresponding OOB data were used to generate an OOB survival curve for each patient. Growing 1000 trees would yield approximately 370 OOB survival curves for each patient and these curves were averaged to yield a risk-adjusted OOB ensemble survival curve to obtain a stable result for this patient. From the curves of patients within a same group, a synthetically risk-adjusted disease failure probability was extracted for this group[26,31]. This curve represented the cumulative probability of disease failure in this group of patients over a period of time. Based on the derivative of the curve, time-specific event-occurring probability for each month was calculated. The processing of RSF methodology was implemented by R version 3.4.3 random Forest SRC package (Version 2.6.1)[32], and the R code used in this study was documented in the Supplementary Software 1.

The current follow-up strategies were recommended by NCCN or RTOG, and were set as control strategies:

1. The most intensive surveillance according to NCCN: every 1 month in year 1; every 2 months in year 2; every 4 months in year 3–5 (27 visits in total);
2. The moderately intensive surveillance according to NCCN: every 2 months in year 1; every 4 months in year 2; every 6 months in year 3–5 (15 visits in total);
3. The least intensive surveillance according to NCCN: every 3 months in year 1; every 6 months in year 2; every 8 months in year 3~5 (10 visits in total);
4. The RTOG strategy: every 3 months during the first 2 years; every 6 months in years 3–5 (14 visits in total).

The most and least intensive surveillance recommended by NCCN was 27 visits and 10 visits in 5 years, respectively. Both the clinical trial RTOG 0225 and 0615 prescribed 14 visits within 5 years after treatment[7,33,34]. Hence, we adjust the total follow-up times of our risk-adjusted surveillance within 5 years ranged from the most 27 visits to the least 5 visits, to explore an optimal surveillance strategy for timely detection of the recurrent disease with the least follow-up times.

Accordingly, we established the risk-based surveillance arrangements with 5-year follow-up times ranged from 5 to 27 visits. And an example was shown in Supplementary Table S1. First, the monthly probability of disease failure (in the second column, labeled as probability per month, in Supplementary Table S1), the total follow-up times (13 for group III in the demonstration) were assigned to each month (in the third column, labeled as follow-ups per month, in Supplementary Table S1). In other words, the more likely it was that the disease failure would recur in that month, the more follow-ups were distributed to it. If the number of follow-ups assigned to a month was less than 0.7, no follow-up was scheduled in that month. The follow-ups in these month(s) for which no visits were scheduled would accrue from month to month until the cumulative follow-up times exceeds 1, when a visit was assigned. Once a visit was scheduled for a given month, the follow-ups began to accumulate again from the next month. If the monthly number of follow-ups exceeded 0.7, a visit was scheduled in that month. A maximum of one visit was scheduled per month, and the excess follow-ups were assigned to other months with high risk. And the details of R code of how the risk-adjusted follow-up time points were scheduled was documented in the Supplementary Software 1.

Then, our risk-based surveillance schedule was compared with the control strategies. Firstly, a hypothetical cohort of 1000 NPC patients with the same features with our study population was generated. And then the effectiveness of surveillance strategies was modeled basing on the sum of delayed-detection months in the hypothetical 1000 patients, which was defined as the duration from the failure occurrence to the next nearest follow-up. For example, if a patient develops distant metastasis in the 6th month while the next nearest planned visit is in the 9th month, the delayed-detection time for this patient is 3 months. The total delayed-detection months of our risk-based surveillance schedule and that of the control strategies were simulated and compared. Strategy using fewer follow-up times and gaining less delayed-detection months was deemed as preferred. The recommendations of risk-based surveillance schedule for patient in different groupings were referring to the ability of moderately intensive NCCN and RTOG surveillance strategy.

**The construction of Markov decision-analytic models.** Markov decision-analytic models were generated to analyze the cost-effectiveness of various follow-up strategies in each patient grouping. In brief, four hypothetical patient cohorts (defined by the grouping mentioned above based on clinical staging and EBV DNA) were evaluated. All patients achieved complete remission after radical radiotherapy or chemoradiotherapy. Starting from the completion of radical treatment, patients were followed up based on the abovementioned strategies. On the recurrence of disease, patients would receive endoscopic nasopharyngectomy or re-irradiation in the event of local recurrence, neck dissection or re-irradiation in the event of neck recurrence, or chemotherapy or chemoradiotherapy in case of distant metastasis. In each Markov model, patients moved through a set of various health states including no evidence of disease, early-stage recurrence, advanced-stage recurrence, salvage treatment for recurrent disease, no evidence of disease after salvage, and death (Supplementary Fig. S3). The observed time horizon was divided into cycles lasting 1 month each.

Parameters including the baseline clinical estimates and utility values required to build the model were derived from published studies[1–9] or, in a few cases, from expert opinions if published data were unavailable, and we used the surveillance and treatment costs reported in 2019 by the Medical Insurance Administration Bureau of Guangzhou, China (Supplementary Table S3).

**Reporting summary.** Further information on research design is available in the Nature Research Reporting Summary linked to this article.

## Data availability

The data of patient characteristics, therapies and survival information have been deposited in the Research Data Deposit public platform (www.researchdata.org.cn, accession code: RDDA2018000934). All the other data of this study are available within the Article, Supplementary Information or from the corresponding author upon reasonable request.

## Code availability

Code and associated instructions with a demo are provided in the Supplementary Software 1 file.

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

## Acknowledgements

The authors would like to thank Prof. Melvin L.K. Chua (from the Division of Radiation Oncology, National Cancer Centre Singapore) for his insightful comments on this study. The authors also thank Yiducloud (Beijing) Technology Ltd. for the establishment of big-data intelligence platform at Sun Yat-Sen University Cancer Centre, and their assistance during the data extraction process. This research was supported by grants from the Pearl River Scholar Funded Scheme, the Guangdong Basic and Applied Basic Research Foundation (2019A1515011863 and 2019A1515012045), the Special Support Program of Sun Yat-sen University Cancer Center (16zxtzlc06), the Health & Medical Collaborative Innovation Project of Guangzhou City, China (201604020003 and 201803040003), the Natural Science Foundation of Guang Dong Province (2017A030312003), the Innovation Team Development Plan of the Ministry of Education (IRT_17R110), Sun Yat-sen University Clinical Research 5010 Program (2012011 and 2014009), the Overseas Expertise Introduction Project for Discipline Innovation (111 Project, B14035), and the National Natural Science Foundation of China (81872463), Key-Area Research and Development Program of Guangdong Province (2019B020230002) and the National Key R&D Program of China(2016YFC0902000).

## Author contributions

Conception and design: Y.S., G-Q.Z., J-h.L. Financial support: Y.S., J.M., G-Q.Z. Administrative support: Y.S., J-h.L., J.M. Provision of study materials or patients: Y.S., J-h.L., G-Q.Z. Collection and assembly of data: G-Q.Z., C-F.W., B.D., T-S.G., J-W.L., L.L., F-p.C., J.K., Z-X.Z., X-D.H., Z-Q.Z. Data analysis and interpretation: G-Q.Z., C-F.W., B.D., T-S.G., J-W.L., L.L., F-p.C., J.K., Z-X.Z., X-D.H., Z-Q.Z., J.M., J-h.L., Y.S. Manuscript writing: All authors. Final approval of manuscript: All authors.

## Competing interests

The authors declare no competing interests.
