## [Peer Review File · Nature Communications]

Reviewers' comments:

Reviewer #1 (Remarks to the Author): Expert in epidemiology

The authors describe a novel approach to surveillance that balances the recurrence risk versus costs.

The topic is of great interest because the optimal post-treatment surveillance strategy is fundamental to detect relapse in a timely manner.

The methodology is widely described, however the authors should explain more clearly why the new strategy is more cost-effective than NCCN strategy. There are no mentions about the costs in the paper.

There are some problems with the symbols I, II etc in the pdf file (pag 14 etc)

Reviewer #2 (Remarks to the Author): Expert in epidemiology and HNSCC

The authors used a random survival forest model to calculate the risk of recurrence for individual NPC patients. Then, based on this risk, a personalised follow-up schedule is determined. The number of clinic visits and the delay in detecting recurrence is determined for the same cohort of subjects for those patients assumed to have used this schedule and those with follow-ups according to the NCCN and RTOG guidelines.

Comments:

1. The arrangement for the follow-up visits according to RSF model is poorly presented in the manuscript. It is difficult to appreciate how the arrangement of individual patient is to be made. Specifically, Figure 3 (which suppose to show the arrangement) is incomprehensible. It is unclear what the meaning of each column and row would be.

2. It is unclear what the difference between the control group and the existing follow-up arrangements which used for generate this set of data is. As the actual time of diagnosis of the recurrence is used as the gold standard, unless the current follow-up arrangement is completely deviated from the existing guidelines, it is unclear why the current standard is shown to have quite significant delay in the diagnosis.

3. The same cohort of patients is used for generating the risk for follow-up schedule and the validation of the method. It is unclear if this would generate overfitting of the data. Ideally, the validation is best performed in an independent cohort of subjects prospectively. With the limitation of time and resources, two independent retrospective cohorts should be used for training and validation.

4. The meaning and significance of Figure 2 is unclear. If I understand correctly, the present model aims to personalised the follow-up schedule of individuals. Based on Figure 2, the risk of recurrence is peaked at certain months. If this is the case, a follow-up schedule with visits at the months with higher risk would also serve the same purpose. However, this may be due to the particular distribution of this cohort of subjects. Hence, an independent cohort is necessary to confirm the finding.

5. It would be useful for a statistician to review the accuracy of the statistical method.

Reviewer #3 (Remarks to the Author): Statistical expert

Zhou et al establish optimal post-treatment surveillance strategy for cancer survivors that balances the recurrence risk versus costs. The surveillance strategy developed is based on a large

database of patients with nasopharyngeal carcinoma (over 6000 patients) treated at their institute and the time-dependent risk score of disease recurrence estimated using machine learning methodology (survival random forest). The time-dependent surveillance strategy developed was evaluated using the sum of delayed-detection months, and compared to the effectiveness of NCCN and Radiation Therapy Oncology Group (RTOG) surveillance guidelines. The advantages of risk-based surveillance strategy were more significant in patients with earlier disease (Figure 4). The study is well conducted. The methodology used to develop the time-dependent, risk-based surveillance strategies in the setting of nasopharyngeal carcinoma could be used for other types of cancer.

Some critiques are as follows:

Major:

1. The authors grouped the patients into 4 risk groups using AJC staging system with additional biomarker (EBV DNA). The time-dependent and risk-based surveillance strategy developed varies according to the 4 risk groups of patients. Other important factors should be part of the risk classification even though the probability of disease failure estimated was adjusted by clinical and genomic factors. For example, supplement Figure 2 shows that age and LDH were as important if not more as T stage which was part of the 4 risk groups classification.
2. Due to the limitation mentioned in #1, the surveillance strategy developed is still suboptimal and the utilization of the strategy may be limited even within the setting of nasopharyngeal carcinoma used for the study.
3. The authors stated 4 endpoints (DF, DMF, LRF, RRF) were assessed. However, the detailed definitions of the 4 time-to-event endpoints was lacking and should be given.
4. It is not exactly clear what risk (e.g. disease-free survival with any kind recurrence as event, local recurrence-free survival, regional recurrence-free survival or distant recurrence-free survival) was used to develop the time-dependent surveillance strategy.
5. Recommendation for risk-based surveillance strategies (Table 2) show the number of visits by year per risk group. It does not provide the timing for each visit.
6. The author should provide the complete R code as the code for fitting survival random forest is missing.

Minor:

1. The detailed cost-effective analysis should be added if relevant data are available even though it is obvious that the cost will be less with a few number of visits under the surveillance strategy developed.
2. Recurrence and relapse were used exchangeable in the manuscript. I would replace relapse by recurrence.
3. Page 178: RSF methodology generates a survival curve for each group of patients, not for each patient.
4. Figure 4: There were no horizontal lines but dots (yellow, red, green) in the figure.
5. In supplemental Figure 1, "%" for count data should be deleted.

Authors' Responses

Reviewer #1 (Expert in epidemiology):

Comment 1. The authors describe a novel approach to surveillance that balances the recurrence risk versus costs.

The topic is of great interest because the optimal post-treatment surveillance strategy is fundamental to detect relapse in a timely manner.

The methodology is widely described, however the authors should explain more clearly why the new strategy is more cost-effective than NCCN strategy. There are no mentions about the costs in the paper.

Response: Thank you for the generous and positive comments on our work. According to your suggestions, we developed Markov decision-analytic models to analyze the cost-effectiveness of various follow-up strategies in each patient group.

In brief, 4 hypothetical patient cohorts (defined by the previously mentioned grouping based on clinical staging and EBV DNA) were evaluated. All patients achieved complete remission (CR) after radical radiotherapy or chemoradiotherapy. Starting from the completion of radical treatment, patients were followed up based on the abovementioned strategies. On the recurrence of disease, patients would receive endoscopic nasopharyngectomy (ENPG) or re-irradiation in the event of local recurrence, neck dissection or re-irradiation in the event of neck recurrence, or chemotherapy or chemoradiotherapy in case of distant metastasis. In each Markov model, patients moved through a set of various health states including no evidence of disease, early-stage recurrence, advanced-stage recurrence, salvage treatment for

recurrent disease, no evidence of disease after salvage, and death (**Supplementary Figure 3**). The observed time horizon was divided into cycles lasting 1 month each.

Supplementary Figure 3. Markov model of health states for patients with nasopharyngeal carcinoma after intensity-modulated radiotherapy. CR, complete remission.

Parameters including the baseline clinical estimates and utility values required to build the model were derived from published studies¹⁻⁹ or, in a few cases, from expert opinions if published data were unavailable, and we used the surveillance and treatment costs reported in 2019 by the Medical Insurance Administration Bureau of Guangzhou, China (**Supplementary Table 3**).

The incremental cost-effectiveness ratios (ICERs) were defined as the incremental cost in US dollars for an additional QALY gained via surveillance strategies compared to the least intensive NCCN surveillance. The baseline cost-effectiveness analyses were summarized in **Table 2**. In general, the risk-based surveillance arrangement was superior to the control strategies in all the groups. For patients in group I, the ICERs for the moderately intensive NCCN surveillance, the most intensive NCCN surveillance, the RTOG strategy and the risk-based strategy were \$2,848/QALY, \$5,072/QALY, \$2,800/QALY, and \$1,957/QALY, respectively, when compared with the least intensive NCCN strategy. Thus, the risk-based strategy had the most favorable ICER. For the remaining patients groups, the risk-based strategy was also the most cost-effective, with ICERs gradually increasing from group II to IV (\$2,407/QALY, \$3,070/QALY and \$3,229/QALY respectively).

We have now included the cost-effectiveness analysis in the revised manuscript (please see **Page 11, Lines 1-17; Page 19, Lines 6-20; Page 20, Lines 1-5**). We thank the reviewer for raising this point; we concur that the additional analyses provide greater comprehensiveness to the interpretation of our data.

Table 2. Baseline cost-effectiveness analysis in Markov models

	Cost (\$)	Incremental cost (\$)	Effectiveness s(QALYs)	Incremental effectiveness	ICER (\$/QALY)
Patients in group □[†]					
The least intensive NCCN strategy	9,187	0	36.049	0	0
The moderately intensive NCCN strategy	11,138	1,951	36.734	0.685	2,848
The most intensive NCCN strategy	15,699	6,512	37.333	1.284	5,072
The RTOG strategy	11,273	2,050	36.780	0.732	2,800
The risk-based strategy [※]	9,372	185	36.142	0.093	1,957
Patients in group □[†]					
The least intensive NCCN strategy	12,479	0	26.627	0	0
The moderately intensive NCCN strategy	15,298	2,819	27.596	0.969	2,909
The most intensive NCCN strategy	19,911	7,432	28.288	1.661	4,474
The RTOG strategy	15,732	3,253	27.734	1.107	2,939
The risk-based strategy [※]	14,869	2,390	27.620	0.993	2,407
Patients in group □[†]					
The least intensive NCCN strategy	14,815	0	21.626	0	0
The moderately intensive NCCN strategy	17,821	3,006	22.514	0.888	3,385
The most intensive NCCN strategy	22,135	7,320	23.040	1.414	5,177
The RTOG strategy	18,333	3,518	22.690	1.064	3,306
The risk-based strategy [※]	17,864	3,049	22.619	0.993	3,070
Patients in group □[†]					
The least intensive NCCN strategy	15,970	0	20.264	0	0
The moderately intensive NCCN strategy	19,111	3,141	21.146	0.882	3,561
The most intensive NCCN strategy	23,367	7,397	21.645	1.381	5,356
The RTOG strategy	19,546	3,576	21.317	1.053	3,396
The risk-based strategy [※]	19,564	3,594	21.377	1.113	3,229

[†] Patients were grouped according to TNM stages and EBV-DNA.

[※]The dominant strategy

Abbreviations: QALY, quality-adjusted life years; ICER, incremental cost-effectiveness ratio; NCCN, National Comprehensive Cancer Network; RTOG, Radiation Therapy Oncology Group.

Comment 2. There are some problems with the symbols I, II etc in the pdf file (pag 14 etc)

Response: We apologize for the inconvenience caused by our carelessness. We have double-checked the system-generated PDF this time and ensured that there were no problems.

References

1. Shen L, *et al.* M1 stage subdivision and treatment outcome of patients with bone-only metastasis of nasopharyngeal carcinoma. *The oncologist* **20**, 291-298 (2015).
2. Zou X, *et al.* Establishment and validation of M1 stage subdivisions for de novo metastatic nasopharyngeal carcinoma to better predict prognosis and guide treatment. *European journal of cancer* **77**, 117-126 (2017).
3. Li YQ, *et al.* Prognostic Model for Stratification of Radioresistant Nasopharynx Carcinoma to Curative Salvage Radiotherapy. *Journal of clinical oncology : official journal of the American Society of Clinical Oncology* **36**, 891-899 (2018).
4. Hua YJ, *et al.* Long-term treatment outcome of recurrent nasopharyngeal carcinoma treated with salvage intensity modulated radiotherapy. *European journal of cancer* **48**, 3422-3428 (2012).
5. Lo WC, *et al.* Salvage treatment for isolated regional failure of nasopharyngeal carcinoma after primary radiotherapy. *Annals of surgical oncology* **19**, 1001-1008 (2012).
6. Wang SY, Lou JL, Chen J, Zhang SZ, Guo L. Salvage surgery for neck residue or recurrence of nasopharyngeal carcinoma after primary radiotherapy: options of surgical methods and regions. *World journal of surgical oncology* **14**, 89 (2016).
7. Hayman J, Weeks J, Mauch P. Economic analyses in health care: an introduction to the methodology with an emphasis on radiation therapy. *International journal of radiation oncology, biology, physics* **35**, 827-841 (1996).
8. Yen RF, Yen MF, Hong RL, Tzen KY, Chien CR, Chen TH. The cost-utility analysis of 18-fluoro-2-deoxyglucose positron emission tomography in the diagnosis of recurrent nasopharyngeal carcinoma. *Academic radiology* **16**, 54-60 (2009).
9. Wu Q, Liao W, Huang J, Zhang P, Zhang N, Li Q. Cost-effectiveness analysis of gemcitabine plus cisplatin versus docetaxel, cisplatin and fluorouracil for induction chemotherapy of locoregionally advanced nasopharyngeal carcinoma. *Oral oncology* **103**, 104588 (2020).

Reviewer #2 (Expert in epidemiology and HNSCC):

The authors used a random survival forest model to calculate the risk of recurrence for individual NPC patients. Then, based on this risk, a personalized follow-up schedule is determined. The number of clinic visits and the delay in detecting recurrence is determined for the same cohort of subjects for those patients assumed to have used this schedule and those with follow-ups according to the NCCN and RTOG guidelines.

Comment 1. The arrangement for the follow-up visits according to RSF model is poorly presented in the manuscript. It is difficult to appreciate how the arrangement of individual patient is to be made. Specifically, Figure 3 (which suppose to show the arrangement) is incomprehensible. It is unclear what the meaning of each column and row would be.

Response: thank you for taking the time to present these perspectives. To clarify this issue, we have included the arrangements for follow-up visits according to the RSF model in detail. An example of the risk-based surveillance arrangements was shown in **Supplementary Table 1**. First, the monthly probability of disease failure (in the second column, labelled “probability per month”, in **Supplementary Table 1**), the total follow-up times (13 for group III in the demonstration) were assigned to each month (in the third column, labelled “follow-ups per month” in **Supplementary Table 1**). In other words, the more likely it was that the disease failure would recur in that month, the more follow-ups were distributed to it. If the number of follow-ups

assigned to a month was less than 0.7, no follow-up was scheduled in that month. The follow-ups in these month(s) for which no visits were scheduled would accrue from month to month until the cumulative follow-up times exceeds 1, when a visit was assigned. Once a visit was scheduled for a given month, the follow-ups began to accumulate again from the next month. If the monthly number of follow-ups exceeded 0.7, a visit was scheduled in that month. A maximum of one visit was scheduled per month, and the excess follow-ups were assigned to other months with high risk. For clarification, we have elaborated the arrangement for the follow-up visits in the Methods section of the revised manuscript (please see **Page 17, Line 16 to Page 18, Line 9**).

Figure 3 shows the risk-based arrangement for different total number of follow-ups (varying from 27 to 5, the abscissa in **Figure 3**) from year 1 to 5 for groups I-IV (the ordinate in **Figure 3**). The follow-up arrangements of a total of 13 visits in patients in group III (4, 4, 3, 1 and 1 visits in years 1-5 respectively) were highlighted in the revised **Figure 3** for clarify. We have also made corresponding revisions in the Results section of the revised manuscript to improve readability (please see **Page 9, Lines 5-9**).

Supplementary Table 1. Follow-up arrangements for a total of 13 visits in the patients of group III

Month	Probability per month	Follow-ups per month	Cumulative follow-ups	Follow-up scheduled
1	0	0	0	0
2	0.005526	0.071841	0.071841	0
3	0.005531	0.071897	0.143738	0
4	0.015512	0.20165	0.345388	0
5	0.011643	0.151363	0.49675	0
6	0.026589	0.345655	0.842405	1
7	0.045163	0.587116	1.429521	0
8	0.03742	0.486464	1.915985	1
9	0.033752	0.438777	2.354762	0
10	0.047574	0.618465	2.973227	1
11	0.031506	0.409576	3.382802	0
12	0.036826	0.478741	3.861543	1
13	0.041297	0.536865	4.398408	0
14	0.025025	0.325328	4.723736	1
15	0.03551	0.461636	5.185372	0
16	0.043128	0.560664	5.746036	1
17	0.021956	0.285426	6.031462	0
18	0.023226	0.301939	6.333401	0
19	0.036375	0.472877	6.806278	1
20	0.028038	0.364488	7.170766	0
21	0.020612	0.26796	7.438725	0
22	0.015779	0.205124	7.643849	0
23	0.017531	0.227904	7.871754	1
24	0.016015	0.208201	8.079955	0
25	0.019207	0.24969	8.329646	0
26	0.017201	0.223614	8.553259	0
27	0.013057	0.169739	8.722999	1
28	0.02016	0.262074	8.985073	0
29	0.02743	0.356595	9.341668	0

30	0.025725	0.334425	9.676093	1
31	0.01417	0.184205	9.860298	0
32	0.01573	0.204484	10.06478	0
33	0.006687	0.086932	10.15171	0
34	0.015404	0.200254	10.35197	0
35	0.011443	0.148765	10.50073	0
36	0.019128	0.248669	10.7494	1
37	0.012111	0.157448	10.90685	0
38	0.013166	0.171159	11.07801	0
39	0.008756	0.113822	11.19183	0
40	0.006459	0.083966	11.2758	0
41	0.00637	0.082813	11.35861	0
42	0.00701	0.09113	11.44974	0
43	0.009001	0.11701	11.56675	0
44	0.009886	0.128515	11.69526	1
45	0.011299	0.146887	11.84215	0
46	0.008599	0.111793	11.95394	0
47	0.003455	0.044921	11.99887	0
48	0.006271	0.081526	12.08039	0
49	0.012101	0.157313	12.2377	0
50	0.003482	0.04526	12.28296	0
51	0.15167	0.15167	12.43464	0
52	0.104245	0.104245	12.53888	1
53	0.046183	0.046183	12.58506	0
54	0.082151	0.082151	12.66721	0
55	0.038068	0.038068	12.70528	0
56	0.119827	0.119827	12.82511	0
57	0.028333	0.028333	12.85344	0
58	0.074934	0.074934	12.92838	0
59	0.031623	0.031623	12.96	0
60	0.04	0.04	13	0
Total	1.0000	13.0000	\	13

Figure 3. The risk-based surveillance arrangements varied 27 to 5 visits for early detection of disease failure, the follow-up arrangements for a total of 13 visits for group III are highlighted.

Comment 2. It is unclear what the difference between the control group and the existing follow-up arrangements which used for generate this set of data is. As the actual time of diagnosis of the recurrence is used as the gold standard, unless the current follow-up arrangement is completely deviated from the existing guidelines, it is unclear why the current standard is shown to have quite significant delay in the diagnosis.

Response: We thank the reviewer for the kind comments. We recognize that the follow-up strategy should be designed to detect recurrence as early as possible. The

ideal situation would be to examine the patient as frequently as possible, daily or weekly. Only in this way would it be possible to detect all recurrence events without any delay. However, it is necessary to balance early detection with surveillance cost by appropriately reducing the number of follow-ups. Therefore, any follow-up strategy will inevitably have some delay. Although our proposed strategy scheduled the follow-up visits in the months with high recurrence risk, delays were inevitable due to the limited number of follow-up visits. The comparison to the control strategies showed that our follow-up arrangement was able to reduce both the delayed-detection time and the number of follow-up visits. The applicability of our strategy was validated in an independent external cohort (please see the response to **Comment 3 & 4** below).

Comment 3. The same cohort of patients is used for generating the risk for follow-up schedule and the validation of the method. It is unclear if this would generate overfitting of the data. Ideally, the validation is best performed in an independent cohort of subjects prospectively. With the limitation of time and resources, two independent retrospective cohorts should be used for training and validation.

& Comment 4. The meaning and significance of Figure 2 is unclear. If I understand correctly, the present model aims to personalised the follow-up schedule of individuals. Based on Figure 2, the risk of recurrence is peaked at certain months. If this is the case, a follow-up schedule with visits at the months with higher risk would also serve the same purpose. However, this may be due to the particular distribution

of this cohort of subjects. Hence, an independent cohort is necessary to confirm the finding.

Response: We appreciate your suggestion. We could not agree more that an independent cohort is necessary to validate the results of our study. Accordingly, we included an external validation cohort from Wuzhou Red Cross Hospital (WZRCH, N=627); and the baseline demographic and disease features were summarized in **Table 1** together with the training cohort from SYSUCC.

Table 1. Demographic and baseline characteristics of patients in the training and validation cohorts

Characteristics	Training cohort (N=6,416)	Validation cohort (N=627)
Age, years		
<45	3,116 (48.6)	203 (32.3)
≥45	3,300 (51.4)	424 (67.7)
Gender		
Male	4,753 (74.1)	446 (71.1)
Female	1,663 (25.9)	181 (28.9)
Smoking		
No	4,107 (64.0)	520 (82.9)
Yes	2,309 (36.0)	107 (17.1)
Alcohol		
No	5,510 (85.9)	549 (87.6)
Yes	906 (14.1)	78 (12.4)
Family history		
No	4,706 (73.3)	603 (96.2)
Yes	1,710 (26.57)	24 (3.8)

Histological type		
WHO Type I	36 (0.6)	25 (4)
WHO Type IIa/IIb	6,380 (99.4)	602 (96)
T category*		
T1	667 (10.4)	96 (15.3)
T2	1,141 (17.8)	146 (23.3)
T3	3,048 (47.5)	141 (22.5)
T4	1,560 (24.3)	244 (38.9)
N category*		
N0	820 (12.8)	43 (6.9)
N1	2,965 (46.2)	326 (52.0)
N2	1,973 (30.8)	170 (27.1)
N3	658 (10.3)	88 (14.0)
EBV DNA		
≤ 2000 copies/mL	2,971 (46.3)	468 (74.6)
> 2000 copies/mL	3,445 (53.7)	159 (25.4)
Groupings [#]		
I	367 (5.7)	22 (3.5)
II	3,472 (54.1)	300 (47.8)
III	1,863 (29.0)	217 (34.6)
IV	714 (11.1)	88 (14.0)
Chemotherapy		
CRT	5,678 (88.5)	589 (93.9)
RT alone	738 (11.5)	38 (6.1)
HGB		
≤ 130 g/L	1,180 (18.4)	270 (43.1)
>130 g/L	5,236 (81.6)	357(56.9)
ALB		
≤ 40 g/L	609 (9.5)	161 (25.7)

>40 g/L	5,807 (90.5)	466 (74.3)
CRP		NA
≤ 3 mg/L	6,515 (69.7)	
> 3 mg/L	2,833 (30.3)	
LDH		NA
≤ 245 IU/L	5,891 (91.8)	
> 245 IU/L	515 (8.2)	

WHO, World Health Organization; EBV, Epstein-Barr virus; HGB, hemoglobin; ALB, albumin; CRP, C-reactive protein; LDH, lactate dehydrogenase; CRT, chemoradiotherapy; RT, radiotherapy; IU, international unit (s).

#Patients were grouped according to T category, N category and EBV DNA

*According to the American Joint Committee on Cancer, 8th edition.

Cost effectiveness analysis in the validation cohort showed that the risk-based strategy was dominant compared with the control strategies in all groups, and the baseline results are summarized in **Supplementary Table 2**. Although the demographic and baseline characteristics of the validation cohort were rather different from those of the training cohort, the risk-based surveillance strategy showed excellent applicability.

The content related to the external validation analysis was added to the Results section (please see **Page 7, Line 3 and Page 11, Lines 12-17**) and the Materials & Methods section (please see **Page 12, Lines 16-17 and Page 12, Line 21 to Page 13, Line 2**).

Supplementary Table 2. Cost-effectiveness analysis in the validation cohort

	Cost (\$)	Incremental cost (\$)	Effectiveness (QALYs)	Incremental effectiveness	ICER (\$/QALY)
Patients in group □[†]					
The least intensive NCCN strategy	7,822	0	36.872	0	0
The moderately intensive NCCN strategy	8,629	807	37.356	0.484	1,667
The most intensive NCCN strategy	12,460	4,638	37.713	0.841	5,515
The RTOG strategy	8,672	850	37.357	0.485	1,753
The risk-based strategy [※]	7,793	-29	36.924	0.052	-558
Patients in group □[†]					
The least intensive NCCN strategy	10,881	0	32.398	0	0
The moderately intensive NCCN strategy	13,491	2,610	33.284	0.886	2,946
The most intensive NCCN strategy	18,520	7,639	34.076	1.678	4,552
The RTOG strategy	13,759	2,878	33.364	0.966	2,979
The risk-based strategy [※]	12,887	2,006	33.167	0.769	2,609
Patients in group □[†]					
The least intensive NCCN strategy	15,168	0	19.201	0	0
The moderately intensive NCCN strategy	17,412	2,244	19.798	0.597	3,759
The most intensive NCCN strategy	20,901	5,733	20.106	0.905	6,335
The RTOG strategy	17,573	2,405	19.904	0.703	3,421
The risk-based strategy [※]	17,174	2,006	19.847	0.646	3,105
Patients in group □[†]					
The least intensive NCCN strategy	17,155	0	17.538	0	0
The moderately intensive NCCN strategy	19,231	2,076	18.052	0.514	4,039
The most intensive NCCN strategy	22,286	5,131	18.282	0.744	6,897
The RTOG strategy	18,908	1,753	18.061	0.523	3,352
The risk-based strategy [※]	18,903	1,748	18.098	0.56	3,121

[†] Patients were grouped according to TNM stages and EBV DNA.

[※]The dominant strategy.

Abbreviations: QALY, quality-adjusted life years; ICER, incremental cost-effectiveness ratio; NCCN, national comprehensive cancer network; RTOG, Radiation Therapy Oncology Group.

Comment 5. It would be useful for a statistician to review the accuracy of the statistical method.

Response: Thanks for your suggestion. A statistical expert has reviewed our study and made some suggestions, and we have revised and added to the manuscript according to his/her suggestions (please see our response to **Reviewer #3** (statistical expert)).

Reviewer #3 (Remarks to the Author): Statistical expert

Zhou et al establish optimal post-treatment surveillance strategy for cancer survivors that balances the recurrence risk versus costs. The surveillance strategy developed is based on a large database of patients with nasopharyngeal carcinoma (over 6000 patients) treated at their institute and the time-dependent risk score of disease recurrence estimated using machine learning methodology (random survival forest). The time-dependent surveillance strategy developed was evaluated using the sum of delayed-detection months, and compared to the effectiveness of NCCN and Radiation Therapy Oncology Group (RTOG) surveillance guidelines. The advantages of risk-based surveillance strategy were more significant in patients with earlier disease (Figure 4). The study is well conducted. The methodology used to develop the time-dependent, risk-based surveillance strategies in the setting of nasopharyngeal carcinoma could be used for other types of cancer.

Some critiques are as follows:

Major:

Comment 1. The authors grouped the patients into 4 risk groups using AJC staging system with additional biomarker (EBV DNA). The time-dependent and risk-based surveillance strategy developed varies according to the 4 risk groups of patients. Other important factors should be part of the risk classification even though the probability of disease failure estimated was adjusted by clinical and genomic factors.

For example, supplement Figure 2 shows that age and LDH were as important if not more as T stage which was part of the 4 risk groups classification.

Response: Thank you for the thoughtful comment, we agree entirely. Several previous studies have revealed the prognostic factors of NPC. Aside from the pre-treatment clinical stage and plasma EBV DNA viral load, many factors have been reported to have prognostic value, such as age, sex, body mass index (BMI), high-sensitivity C-reactive protein (hs-CRP), lactate dehydrogenase (LDH), and hemoglobin (HGB)¹⁻⁴. In fact, we initially attempted to take all the reported prognostic factors into account for patient grouping. However, we found that the grouping results were too complicated to present and would reduce the applicability of the proposed follow-up strategy. Therefore, in the end, we included only the two factors currently recognized as most well-recognized in NPC and widely-applied in different institution worldwide: clinical stage and EBV DNA. First of all, clinical staging has always been the key determinant for prognostic prediction and treatment decisions in NPC⁵. On the other hand, plasma EBV DNA, which is now gradually being adopted in clinical applications, is currently considered as the most attractive potential biomarker to complement the clinical staging⁶⁻⁹.

In a previous study from our group, we proposed TNM stage groupings incorporating EBV DNA, and the area under the ROC curve for PFS was as high as 0.69¹⁰, which was slightly inferior to the prognostic performance of a monogram integrating all the prognostic indicators (the C-index was 0.728)¹¹. Therefore, to make the grouping simple and convenient for clinical use, we used the simpler

grouping method and included only the clinical stage and EBV DNA as patient grouping factors. As an alternative, we entered the other factors into the random survival forest model for risk adjustment. The result from the external validation cohort showed that this model had excellent adaptability. Although the demographic and baseline characteristics of the validation cohort were rather different from those of the training cohort, the risk-based surveillance strategy was more cost-effective than the control strategies. Thank you again for pointing this out. We have added this point to the Limitations section of the revised manuscript (please see **Page 23, Lines 3 to 5**).

Comment 2. Due to the limitation mentioned in #1, the surveillance strategy developed is still suboptimal and the utilization of the strategy may be limited even within the setting of nasopharyngeal carcinoma used for the study.

Response: Thanks for your comments. According to the suggestions from the reviewers, we have included an external cohort for validation, which included 627 NPC from Wuzhou Red Cross Hospital (WZRCH). The cost effectiveness analysis of the validation cohort verified that the risk based surveillance strategy was the dominant compared with the control strategies in all groups. Although the demographic and baseline characteristics of the validation cohort were rather different from those of the training cohort, the risk-based strategy had excellent applicability and remained the most cost effective (please see **Page 12, Lines 16-17 and Page 12, Line 21 to Page 13, Line 2**).

Because the relapse site and time of other tumours are different from those of NPC, the surveillance strategy generated according to the recurrence characteristics of NPC, the surveillance strategy generated according to the recurrence characteristics of NPC is definitely not suitable for other tumours. Although this model was established in the context of NPC, our method of modelling risk-based surveillance was universally applicable for the development of cost-effective surveillance strategies, and would assist in shaping individualized, risk-based post-treatment follow-up for the cancer survivors in general.

Comment 3. The authors stated 4 endpoints (DF, DMF, LRF, RRF) were assessed. However, the detailed definitions of the 4 time-to-event endpoints was lacking and should be given.

Response: Thank you for your suggestion. We have added the detailed definitions of DF, DMF, LRF and RRF to the revised text: DF was calculated from the date of diagnosis to the date of local or regional recurrence, distant metastasis, or death from any cause, whichever occurred first. DMF, LRF and RRF were calculated from the date of diagnosis to the dates of distant metastasis, local recurrence and regional recurrence, respectively (please see **Page 15, Lines 15 to 18**).

Comment 4. It is not exactly clear what risk (e.g. disease-free survival with any kind recurrence as event, local recurrence-free survival, regional recurrence-free survival or distant recurrence-free survival) was used to develop the time-dependent surveillance strategy.

Response: We apologize for not making this information clear in the manuscript. In this study, we use disease-free survival with any type of recurrence event, including local recurrence, regional recurrence and distant metastasis, to develop the time-dependent surveillance strategy. We have clarified this in the revised manuscript (please see **Page 15, Lines 5 to 6**).

Comment 5. Recommendation for risk-based surveillance strategies (Table 2) show the number of visits by year per risk group. It does not provide the timing for each visit. 5

Response: Thank you for the kind reminder. We have changed the table to **Figure 5** to more intuitively show the timing of each visit.

Figure 5. Recommended risk-based surveillance arrangements for patients in each group from year 1-5; the darkened boxes of the grid represent the months recommended for visits.

Comment 6. The author should provide the complete R code as the code for fitting survival random forest is missing.

Response: All the R code used in this study is listed in a README file in the supplementary data.

Minor:

Comment 7. The detailed cost-effective analysis should be added if relevant data are available even though it is obvious that the cost will be less with a few number of visits under the surveillance strategy developed.

Response: Your suggestion is very important. We developed Markov decision-analytic models to analyse the cost-effectiveness of various follow-up strategies in each patient grouping. The analysis showed that the risk-based surveillance arrangement was superior to the control strategies in all the groups, and this result was verified in the independent validation group. The details of the cost-effectiveness analysis are described in the manuscript (please see **Page 11, Lines 1-17; Page 19, Lines 6-20; Page 20, Lines 1-5**).

Comment 8. Recurrence and relapse were used exchangeable in the manuscript. I would replace relapse by recurrence.

Response: Thanks you for your suggestion. We have replaced the term “relapse” with “recurrence” throughout the text.

Comment 9. Page 178: RSF methodology generates a survival curve for each group of patients, not for each patient.

Response: We have revised the manuscript according to your suggestion (please see **Page 16, Line 2**).

Comment 10. Figure 4: There were no horizontal lines but dots (yellow, red, green) in the figure.

Response: As suggested, we have added the horizontal lines for the dots in **Figure 4**.

Figure 4. Delays in the detection of disease failure in risk-based surveillance arrangements (blue curve) compared with the control follow-up strategies (the yellow, green and blue horizontal lines, respectively, represent the most intensive, moderately intensive and least intensive surveillance strategies according to NCCN; the brown horizontal line represents the RTOG strategy) for patients in group I (A), group II (B), group III (C) and group IV (D).

Comment 11. In supplemental Figure 1, “%” for count data should be deleted.

Response: Thank you for your reminder. We have deleted “%” for count data in **Supplementary Figure 1.**

Supplementary Figure 1. The number of disease failure events (on the left) and the crude incidence (on the right) month by month in the different patient groups.

References

1. Zhou GQ, *et al.* Baseline serum lactate dehydrogenase levels for patients treated with intensity-modulated radiotherapy for nasopharyngeal carcinoma: a predictor of poor prognosis and subsequent liver metastasis. *International journal of radiation oncology, biology, physics* **82**, e359-365 (2012).
2. Xia WX, *et al.* A prognostic model predicts the risk of distant metastasis and death for patients with nasopharyngeal carcinoma based on pre-treatment serum C-reactive protein and N-classification. *European journal of cancer* **49**, 2152-2160 (2013).
3. Lin YH, Chang KP, Lin YS, Chang TS. Evaluation of effect of body mass index

and weight loss on survival of patients with nasopharyngeal carcinoma treated with intensity-modulated radiation therapy. *Radiation oncology* **10**, 136 (2015).

4. Zhang LN, Tang J, Lan XW, OuYang PY, Xie FY. Pretreatment anemia and survival in nasopharyngeal carcinoma. *Tumour biology : the journal of the International Society for Oncodevelopmental Biology and Medicine* **37**, 2225-2231 (2016).
5. Lee AW, Ma BB, Ng WT, Chan AT. Management of Nasopharyngeal Carcinoma: Current Practice and Future Perspective. *Journal of clinical oncology : official journal of the American Society of Clinical Oncology* **33**, 3356-3364 (2015).
6. Song C, Yang S. A meta-analysis on the EBV DNA and VCA-IgA in diagnosis of Nasopharyngeal Carcinoma. *Pakistan journal of medical sciences* **29**, 885-890 (2013).
7. Ma BB, *et al.* Relationship between pretreatment level of plasma Epstein-Barr virus DNA, tumor burden, and metabolic activity in advanced nasopharyngeal carcinoma. *International journal of radiation oncology, biology, physics* **66**, 714-720 (2006).
8. Lo YM, *et al.* Plasma cell-free Epstein-Barr virus DNA quantitation in patients with nasopharyngeal carcinoma. Correlation with clinical staging. *Annals of the New York Academy of Sciences* **906**, 99-101 (2000).
9. Leung SF, *et al.* Plasma Epstein-Barr viral deoxyribonucleic acid quantitation complements tumor-node-metastasis staging prognostication in nasopharyngeal carcinoma. *Journal of clinical oncology : official journal of the American Society of Clinical Oncology* **24**, 5414-5418 (2006).
10. Guo R, *et al.* Proposed modifications and incorporation of plasma Epstein-Barr virus DNA improve the TNM staging system for Epstein-Barr virus-related nasopharyngeal carcinoma. *Cancer* **125**, 79-89 (2019).
11. Tang LQ, *et al.* Establishment and Validation of Prognostic Nomograms for Endemic Nasopharyngeal Carcinoma. *Journal of the National Cancer Institute* **108**, (2016).

Reviewer #2 (Remarks to the Author):

The authors have provided detailed answers to my previous queries.
The inclusion of an independent cohort has provided important validation data to the proposed method.

All my concerns have been adequately addressed.

Reviewer #3 (Remarks to the Author):

The authors were responsive to the critiques and addressed them with satisfaction and additional analysis. No further substantial comments.